# Fermentation Dynamics of Naturally Fermented Palm Beverages of West Bengal and Jharkhand in India

**Souvik Das and Jyoti Prakash Tamang ***

Department of Microbiology, School of Life Sciences, Sikkim University, Gangtok 737102, Sikkim, India
* Correspondence: jptamang@cus.ac.in; Tel.: +91-983-206-1073

**Abstract:** The term '*toddy*' represents a group of different varieties of mild-alcoholic palm beverages of coastal and inland India, produced from the fresh saps of various palm trees through uncontrolled natural fermentation. In this study, we analysed the successional changes of microbial abundances and various physico-chemical parameters during natural fermentation (0 h to 48 h) of *taal toddy*, prepared from Palmyra palm, and *khejur toddy*, prepared from date palm of West Bengal and Jharkhand in India. Microorganisms from different successional levels were isolated and grouped using repetitive element sequence-based PCR (rep-PCR) technique and identified by the sequencing of 16S rRNA gene and D1-D2 region of 26S rRNA gene for bacteria and yeasts, respectively. *Enterococcus faecalis*, *Lactiplantibacillus plantarum*, *Lacticaseibacillus paracasei* and yeast *Saccharomyces cerevisiae* were identified during natural fermentation of *toddy*. During the natural fermentation, the average pH and total sugar content in the samples of both *taal* and *khejur toddy* decreased, whereas a gradual rise was observed in the contents of acidity, total alcohol, total ester and total protein. Bio-active potential (presence of phenolics and flavonoids) of *toddy* was also analysed (0 h to 48 h), where contents of total phenolics, flavonoids and resulting anti-oxidant activity were found higher in the end-product than the fresh palm sap, indicating *toddy* as a functional low-alcoholic drink. Lastly, it can be concluded that the inter-variable dynamics and microbial interrelation, which in turn depend on a number of local factors, regulate the overall fermentation dynamics and determine the product quality.

**Keywords:** *toddy*; fermented palm; fermentation dynamics; *Lactiplantibacillus plantarum*; *Saccharomyces cerevisiae*; ethanol

## 1. Introduction

India has a unique dietary culture with a vast diversity of ethnic fermented foods and alcoholic beverages, which are the indispensable components of Indian gastronomy [1]. Besides flavoursome fermented foods, drinking alcoholic fermented beverages as one of the dietary items is a customary provision for many ethnic Indian people [2]. *Toddy* is a milky, effervescent and mild-alcoholic beverage produced by natural fermentation of fresh palm saps, mostly in palm-tree-growing regions in coastal as well as inland India [3,4]. Apart from the Indian sub-continent, the production and consumption of fermented palm beverages are also common in South East Asia, Africa and Latin America [5,6]. Though fermented palm drink is consumed as a refreshing drink in many places, this beverage has also deep socio-cultural values among the different ethnic communities [7,8]. During the natural fermentation of palm sap, a multifarious range of microbiota from the surrounding environment affect the quality and aroma of the product by variegating a vast array of metabolites [9]. Moreover, the colossal diversity of the participating microbes is determined by a number of factors, such as the tapping process, species of the palm plant, geographical variation and agro-climatic factors [10]. Interestingly, the dynamics of fermentation of palm sap are maintained by a stable balance between the physico-chemical parameters and the abundance of microbial communities, where both factors regulate and control each other throughout the fermentation [11,12].

Several species of palm trees, such as Palmyra palm (*Borassus flabellifer* L.), locally called *taal* and silver/wild date palm (*Phoenix sylvestris* Roxb.), known as *khejur*, are grown in Eastern regions of India mostly in states of West Bengal and Jharkhand. The ethnic communities of these regions prepare *toddy* from the fresh saps of locally grown date palm and Palmyra palm trees using their indigenous knowledge of natural fermentation (Figure 1) for consumption and also for their livelihood. During preparation, *toddy* producer climbs on the palm tree, makes a tiny hole in the upper portion of the palm trunk, inserts a bamboo-made hollow pipe into the cut portion of the trunk and the other end of the pipe is inserted into a small, round-shaped earthen pot to collect the fresh sap drop by drop. After collection, the sap is filtered through a strainer to remove the visible impurities and is poured into a clean jar, which is kept outside at room temperature for natural fermentation for 2–3 days to obtain *toddy* (Figure 1). Based on the sensory properties, *taal toddy* has a strong flavour with a milky-white colour, whereas a sweet-fruity aroma is the distinctive feature of *khejur toddy*.

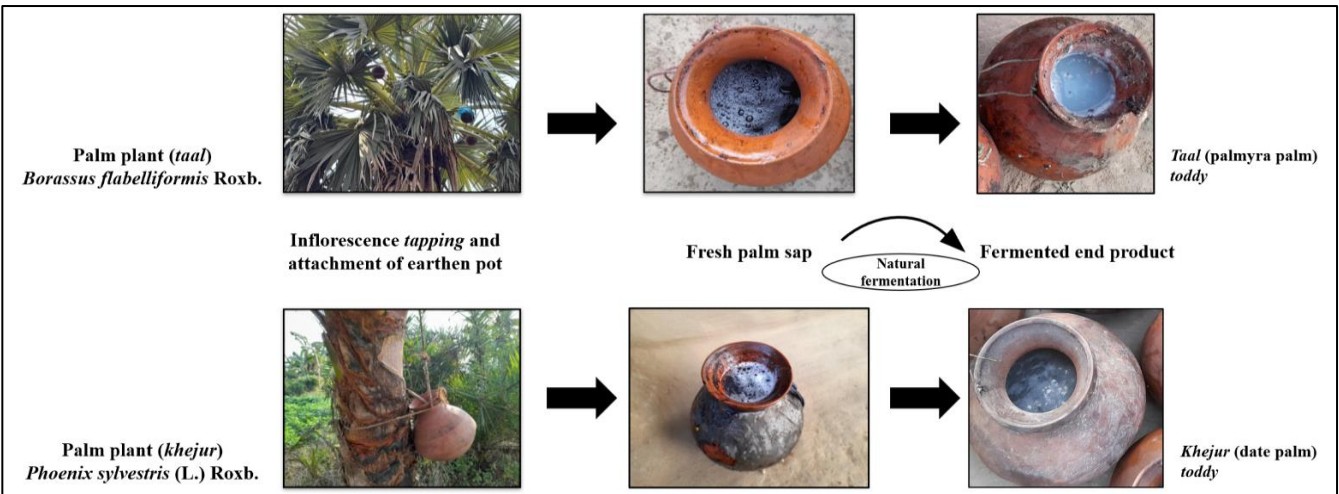

**Figure 1.** Methods of preparation of Indian palm beverage: *taal toddy* (**top**) and *khejur toddy* (**bottom**).

Palm sap and palm beverage are used as a folklore medicine in rural India, which confers a number of healing benefits, such as improvement of eyesight and gastrointestinal health; moreover, this beverage is consumed without any further purification and filtration [6]. So, it is quite important to study the natural fermentation dynamics of this beverage to decipher the product characteristics, which is directly associated with the overall product quality and consumer safety. Studies on microbiological and biochemical aspects of different types of fermented palm beverages have been conducted earlier [4,6,9,11–15]. However, information on fermentation dynamics on inter-variable and intra-variable balances in the natural fermentations of fresh saps of date palm and Palmyra palm of West Bengal and Jharkhand in the Eastern regions of India is unknown. We also hypothesise that the dynamics of natural and uncontrolled fermentation of palm sap are regulated by a stable balance among the different physico-chemical variables, which ultimately characterise the final product. Moreover, these variables also depend on a number of factors, which have already been described earlier in this paper. Hence, this paper is aimed to study the fermentation dynamics of *toddy*, prepared from fresh saps of Palmyra palm known as *taal toddy* and date palm known as *khejur toddy* of Eastern India, focusing on the microbial and physico-chemical changes during the natural fermentation.

## 2. Materials and Methods

### 2.1. Collection of Samples

Three sets of samples from each stage of fermentation of fresh palm saps of Palmyra palm (*Borassus flabellifer* L.) for *taal toddy* were collected aseptically at the interval of 0 h,

24 h and 48 h during natural fermentation from Asanpur village (23.09958 N/88.443374 E; elevation: 17 m) in Hooghly district, West Bengal. Similarly, three sets of samples (from a single earthen pot) from each stage of fermentation of fresh palm saps of date palm (*Phoenix sylvestris* Roxb.) for *khejur toddy* were collected in the sterile containers by maintaining all the necessary precautions to avoid any unwanted contamination, at the intervals of 0 h, 24 h and 48 h during natural fermentation from Poradihi village (23.59011 N/86.45890 E; elevation: 166 m) in Purulia district of West Bengal and Bhojudihi village (23.64118 N/86.44649 E; elevation: 154 m) in Bokaro district of Jharkhand, respectively. *Taal toddy* samples were collected during the summer time (i.e., March–April), whereas *khejur toddy* samples were collected in the period of December–January (winter season). The samples were collected in the aseptic sample containers that were made sterile using the autoclave and temporarily stored at −20 °C in cold storage at a local facility in towns nearby the collection sites, where prior arrangements were made. Once the whole sampling was completed, the samples kept in an ice-box carrier were immediately transported to the Department of Microbiology, Sikkim University, Gangtok, and stored at −20 °C for further analyses. Samples in this study were designated as Hooghly, Purulia and Bokaro according to the districts of collection sites. All the following physico-chemical parameters (in triplicates) were analysed in the samples from each stage/succession of fermentation, viz. 0 h, 24 h and 48 h.

### 2.2. pH, Total Acidity and Total Alcohol Content

The pH of the samples was determined using the digital pH meter (Orion 3-Star, Thermo Fisher Scientific, Waltham, MA, USA). The method of FSSAI (Food Safety and Standards Authority of India) [16] was followed for the estimation of total acidity content in samples with slight modifications. Briefly, the acidity content was analysed by titrating the samples against 0.05 N NaOH and expressed as g/100 mL of lactic acid. The total alcohol content in *toddy* samples was analysed through the dichromate oxidation technique as per the method of AOAC (Association of Official Analytical Collaboration) [17] and expressed as % alcohol (*v/v*).

### 2.3. Proportion Distribution between Ethanol and Methanol

Ethanolic and methanolic proportions in the total alcohol content were calculated using the Fourier transform infrared (FTIR) spectroscopy (Alpha FTIR, Bruker, Germany), as described by Coldea et al. [18]. Different concentrations of pure ethanol and pure ethanol (0–100%) were run first in the FTIR spectrophotometer and the absorbance corresponding to the signature peaks for both alcohols was recorded. The samples were run in the same manner and the absorbance for the signature peaks was compared to the signature peaks of the standard concentrations of pure ethanol and methanol to calculate the proportional difference between them.

### 2.4. °BRIX (Degrees BRIX) Profile and Total Sugar Content

The degrees BRIX (total soluble solid) profile of both the fresh sap and fermented sap samples was determined using the digital hand refractometer (RHB-32 ATC, Erma Inc., Tokyo, Japan), as described by Ofoedu et al. [19]. Initially, the refractometer was standardised at BRIX value zero using sterile distilled water. Subsequently, two drops of wine sample were placed on the light plate/lens and the measurement was recorded. Total sugar content was measured by following the anthrone method of carbohydrate estimation by Pandeya et al. [20]. Anthrone solution was prepared by dissolving 2 g of anthrone in 1 L of conc. $H_2SO_4$. Different concentrations (0–50%) of glucose solution were used to prepare the standard calibration curve. Ultimately, 1 mL of each sample was mixed with 5 mL of anthrone reagent, followed by incubation in a boiling water bath for 10 min. The optical density (OD) was measured at 620 nm and the total sugar content was calculated by putting the OD value on the standard calibration curve.

### 2.5. Ester Content

The total ester content of the samples was measured as per the method of FSSAI [16] with slight modifications and expressed as g/L of absolute alcohol. Briefly, 50 mL of the sample was mixed with 10 mL of 0.1 N NaOH and the mixture was allowed to reflux on a steam bath for 1 h. After cooling, the unspent alkali was back-titrated against 0.1 N $H_2SO_4$. In a similar way, a blank titration was also carried out with distilled water instead of the sample. Finally, the amount of equivalent ester was calculated from the difference in titre value.

### 2.6. Total Crude Protein Content

The total crude protein content of the samples was determined by the Folin-Lowry method as described by PotdarVrushali et al. [21] and expressed as mg/L. Different concentrations of standard bovine serum albumin (BSA) (MB083, HiMedia, Mumbai, India) solution are used to make the calibration curve. Briefly, 5 mL of alkaline CuSO4 was added to 1 mL of samples, distilled water (as blank) and different concentrations of standard BSA in separate test tubes and were mixed thoroughly. The 0.5 mL of freshly prepared Folin's reagent (FC) (39520, SRL, Mumbai, India) was added then to each of the tubes. The tubes were kept at room temperature for 30 min, and the OD was measured at 660 nm. The total protein content of the samples was estimated using the reference of standard BSA concentrations.

### 2.7. Total Phenolics and Flavonoids Contents

Total phenolics and flavonoids in samples were measured using spectrophotometric assay as per the method of Milinčić et al. [22]. Sample (70 μL) was mixed with 300 μL of FC reagent (39520, SRL, Mumbai, India) and 230 μL of 7.5% $Na_2CO_3$ and the volume was made up to 1000 μL using distilled water. The mixture was incubated at room temperature for 2 h and the absorbance was measured at 765 nm using the UV-Vis spectrophotometer (BioSpectrometer Basic, Eppendorf, Hamburg, Germany). The same FC-based method was performed with gallic acid (50–900 mg/L) to plot the standard curve, and the total phenolic content was expressed as gallic acid equivalent (GAE mg/L). Total flavonoid was also determined using a spectrophotometric assay based on the formation of a flavonoid–aluminium complex. Briefly, 125 μL of the sample, 625 μL of ultra-purified water and 37.5 μL of 5% $NaNO_2$ were mixed first. Subsequently, 75 μL of 10% $AlCl_3$ was added to the mixture after 6 min and the mixture was allowed for incubation for 5 min followed by the addition of 250 μL of 1 M NaOH. The absorbance was measured at 510 nm and expressed as quercetin equivalent (mg/L), which was used for the standard calibration curve (50–800 mg/L).

### 2.8. Anti-Oxidant Activity

Anti-oxidant property of samples was measured as per the method described by El Euch et al. [23]. The 0.2 mM 2,2-diphenyl-1-picrylhydrazyl (DPPH) (RM2798, HiMedia, Mumbai, India) solution was prepared first by dissolving 2,2-diphenyl-1-picrylhydrazyl (HiMedia, Mumbai, India) in ice-cold methanol. Then, 900 μL of DPPH solution was added to each of the samples followed by incubation at dark for 30 min. Absorbance was measured at 524 nm against the blank; the blank was considered as the control that contained all the reagents and distilled water, instead of the sample. The following formula was applied to calculate the radical scavenging activity:

$$\text{DPPH radical scavenging activity (\%)} = (A_{Blank} - A_{Sample}) / A_{Blank} \times 100$$

### 2.9. Microbiological Analysis

One mL of the sample was mixed with 9 mL of physiological saline (0.85%) and homogenised thoroughly using a Stomacher blender (Stomacher 400 Circulator Lab Blender, Seward, UK) for 2 min. Homogenised samples were diluted serially in the same diluent;

1 mL of diluent from the particular dilution was then plated on plate count agar (M091S, HiMedia, Mumbai, India) and Yeast Malt (YM) Agar (M424, HiMedia, Mumbai, India) for the enumeration of bacteria and yeasts, respectively. Bacterial plates were incubated at 37 °C for 24 h and yeast plates were incubated at 28 °C for 48 h. After incubation, the number of colonies from the respective plates was counted and expressed as colony-forming units (cfu)/mL.

Lactic acid bacteria (LAB) from samples were enumerated using MRS agar medium (GM641, HiMedia, Mumbai, India) as per the method described by Shangpliang and Tamang [24]. Homogenised samples were inoculated into MRS agar plates supplemented with 1% $CaCO_3$ by pour plating method. The plates were incubated anaerobically at 30 °C for 48 h. After incubation, isolates were selected primarily and re-inoculated into the MRS broth (M1926, HiMedia, Mumbai, India) and incubated for 24 h at 30 °C and their growth was checked at 600 nm using a spectrophotometer. Based on the growth ($A_{600} \geq 1.00$), isolates were again selected finally for further analyses.

Yeasts from samples were enumerated following the method described by Lama and Tamang [25]. Homogenised samples were inoculated into YM agar plates followed by incubation for 48 h at 28 °C. Isolates were screened primarily by flocculation test and ethanol tolerance test (100 mL/L, 130 mL/L and 150 mL/L) as described by Guimarães et al. [26].

*2.10. Genomic DNA Isolation*

Genomic DNA of LAB isolates was extracted as per the method of Shangpliang and Tamang [24]. Briefly, 2 mL of bacterial culture was transferred into a micro-centrifuge tube and centrifuged at 8000× *g* for 5 min at 4 °C. The supernatant was discarded and the remaining pellet was washed twice with 0.5 M NaCl, followed by another round of washing with sterile nuclease-free water. Then, the pellet was suspended in 1X TE (Tris-EDTA) buffer (ML060, HiMedia, Mumbai, India) and 10 µL of lysozyme (RM074, HiMedia, Mumbai, India) with a concentration of 2 mg/mL, followed by a short period of incubation for 30 min at 37 °C. The suspension was then kept at 98 °C for 15 min and then centrifuged at 10,000× *g* for 10 min. Finally, the supernatant was transferred into a fresh tube and the DNA was checked using 1% agarose gel electrophoresis. The DNA was quantified and the purity was checked using BioSpectrometer (BioSpectrometer Basic, Eppendorf, Hamburg, Germany).

The method of Lama and Tamang [25] was followed for the isolation of DNA from yeast isolates. Overnight grown culture was taken into a micro-centrifuged tube and centrifuged at 12,000× *g* for 10 min. After discarding the supernatant, 400 µL of lysis buffer (Tris-HCL pH 8.0, 0.5 M EDTA pH 8.0, 5 M NaCl, 10% SDS) and 2 µL of RNase A (97610, SRL, Mumbai, India) were added with the remaining pellet and the suspension was incubated at 65 °C for 30 min. Subsequently, after the incubation, 5 µL of Proteinase K (RM2957, HiMedia, Mumbai, India) was added with the suspension and again incubated at 65 °C for 30 min. The 100 µL of 5 M NaCl was added and the suspension was re-incubated at −20 °C for 10 min. The suspension was centrifuged then at 12,000× *g* for 10 min and the supernatant was transferred into a fresh tube, after which an equal volume of PCI mixture (phenol–chloroform–isoamyl alcohol) (25:24:1 *v/v*) (69031, SRL, Mumbai, India) was added. The suspension was re-centrifuged at 12,000× *g* for 10 min and the upper aqueous layer was carefully transferred into a new tube, mixed with ice-cold isopropanol and kept for overnight incubation at −20 °C. Then, it was centrifuged again at 14,000× *g* for 10 min followed by the removal of supernatant and addition of 100 µL of chilled 70% alcohol and finally re-centrifuged at 8000× *g* for 5 min. The supernatant was discarded and the pellet was allowed for air-drying. The pellet was dissolved in 50 µL of sterilised nuclease-free water. Prior to further analyses, the DNA was quantified using nanodrop (BioSpectrometer Basic, Eppendorf, Hamburg, Germany) and checked through 1% agarose gel electrophoresis.

### 2.11. Repetitive Sequence-Based PCR

LAB and yeasts isolated were grouped through the repetitive sequence-based PCR using $GTG_5$ primer (5′-GTGGTGGTGGTGGTG-3′) as described by Cissé et al. [27]. The PCR profile was initially denatured at 94 °C for 4 min, followed by 30 cycles of denaturation at 94 °C for 30 s, annealing at 45 °C for 1 min, elongation at 65 °C for 8 min and finally extended at 65 °C for 16 min with cooling at 4 °C for infinite time. Isolates were then grouped based on their banding patterns. One representative isolate from each of the bacterial and yeast groups was chosen for further amplification and sequencing. While selecting the representative isolate, previously mentioned criteria (growth for bacteria, and flocculation and ethanol tolerance for yeast) were taken into consideration.

### 2.12. Amplification and DNA Sequencing

The 16S rRNA gene from the representative bacterial isolates was amplified using 27F (27F 5′-AGAGTTTGATCATGGCTCAG-3′) and 1492R (5′-GTTACCTTGTTACGACTT-3′) primers by following the method of Shangpliang and Tamang [24]. The PCR reaction mixture (25 μL) contained the required amount of all four dNTPs, $MgCl_2$, polymerase enzyme (in the form of PCR master mix, M7122, Promega, Madison, WI, USA), both the primers and 30–50 ng of template DNA. The PCR amplification was performed using VeritiPro thermal cycler (Thermo Fisher Scientific, Waltham, MA, USA) with the following profile: initial denaturation at 94 °C for 5 min, followed by the 30 cycles of denaturation at 94 °C for 1 min, annealing at 55 °C for 1 min, elongation at 72 °C for 1.5 min and the final elongation at 72 °C for 10 min, followed by cooling at 4 °C for infinite time.

For representative yeast isolates, the D1–D2 region of the 26S rRNA gene was amplified for the final identification using NL1 (5′-GCA TAT CAA TAA GCG GAG GAA AAG-3′) and NL4 (5′-GGT CCG TGTTTC AAG ACG G-3′) primers as described by Palla et al. [28]. The volume of the PCR reaction mixture was kept the same as bacteria (25 μL), and it contained all four dNTPs, $MgCl_2$, polymerase enzyme (in the form of PCR master mix, M7122, Promega, Madison, WI, USA), both the primers and 10–20 ng of template DNA. The PCR amplification was performed using VeritiPro thermal cycler (Thermo Fisher Scientific, Waltham, MA, USA) with the following profile: initial denaturation at 94 °C for 1 min, followed by the 35 cycles of denaturation at 94 °C for 30 s, annealing at 58 °C for 30 s, elongation at 72 °C for 30 s and the final elongation at 72 °C for 5 min, followed by cooling at 4 °C for infinite time.

The amplicons for both the bacterial and yeast isolates were checked using 1% agarose gel electrophoresis and visualised using Gel Doc™ EZ (Bio-Rad, Richmond, CA, USA) prior to the sequencing using an automated DNA analyser (3500 Genetic Analyzer, Applied Biosystems, Waltham, MA, USA).

### 2.13. Bioinformatics Analysis

The quality of the raw sequences was checked first using the Sequence Scanner software version 2.0 (https://www.thermofisher.com/in/en/home/life-science/sequencing/sanger-sequencing/sanger-dna-sequencing/sanger-sequencing-data-analysis.html, Applied Biosystems, Waltham, MA, USA) (accessed on 5 December 2022). Sequences with a Q score of $\geq$20–30 were processed for further analyses. Good quality sequences were then assembled to form the contigs in ChromasPro software version 1.34 (http://technelysium.com.au/wp/chromas/) (accessed on 5 December 2022). Mallard was used to check the presence of any type of chimeric sequences. The taxonomic identity of the chimaera-free sequences was then detected through the Basic Local Alignment Search Tool (BLAST) algorithm (https://blast.ncbi.nlm.nih.gov/Blast.cgi) (accessed on 5 December 2022) using the reference rRNA/ITS databases of the National Center for Biotechnology Information (NCBI). Specifically, 16S rRNA database sequences were used as references for bacterial isolates and large-subunit (LSU) reference database sequences of fungi type were used for the identification of yeasts. The phylogenetic relationship was depicted through the phylogenetic tree constructed using MEGA11 software [29], based on the neighbour-joining

method. Prior to the formation of the phylogenetic tree, sequences were aligned with the ClustalW [30] algorithm within the previously mentioned MEGA11 software.

### 2.14. Statistics

All analyses were performed in triplicates and represented in mean ± SD. Statistical variations were calculated through *T*-test using PAST software version 4.0 [31].

## 3. Result

### 3.1. Microbial Changes

The populations of bacteria and yeasts increased gradually during the fermentation of fresh palm saps in all successional samples collected from Hooghly, Purulia and Bokaro. Bacterial count in Hooghly samples showed its increase from $0.81 \times 10^6$ cfu/mL at 0 h of fermentation (fresh sap) to $1.52 \times 10^6$ cfu/mL at the end (48 h of fermentation) with the intermediate load of $1.09 \times 10^6$ cfu/mL at 24 h of fermentation (Figure 2a); however, the yeast population in the same samples increased from $0.67 \times 10^6$ cfu/mL in fresh sap to $1.52 \times 10^6$ cfu/mL in the end product at 48 h (Figure 2b). A steady increase was also observed in Purulia samples, where the bacterial abundance increased from $0.55 \times 10^6$ cfu/mL at the initial stage of fermentation (0 h) to $1.30 \times 10^6$ cfu/mL and $1.44 \times 10^6$ cfu/mL at mid- (24 h) and end (48 h) phases of fermentation, respectively (Figure 2a); yeast populations also expanded their abundances in each successional level during *toddy* fermentation, ranging from $0.42 \times 10^6$ cfu/mL at 0 h to $1.2 \times 10^6$ at the end (48 h) (Figure 2b). Samples from Bokaro showed an increasing bacterial abundance with a load of $0.73 \times 10^6$ cfu/mL and $1.31 \times 10^6$ cfu/mL and yeasts with counts of $1.48 \times 10^6$ cfu/mL and $0.36 \times 10^6$ cfu/mL, $0.98 \times 10^6$ cfu/mL and $1.23 \times 10^6$ cfu/mL, respectively, during fermentation (Figure 2b).

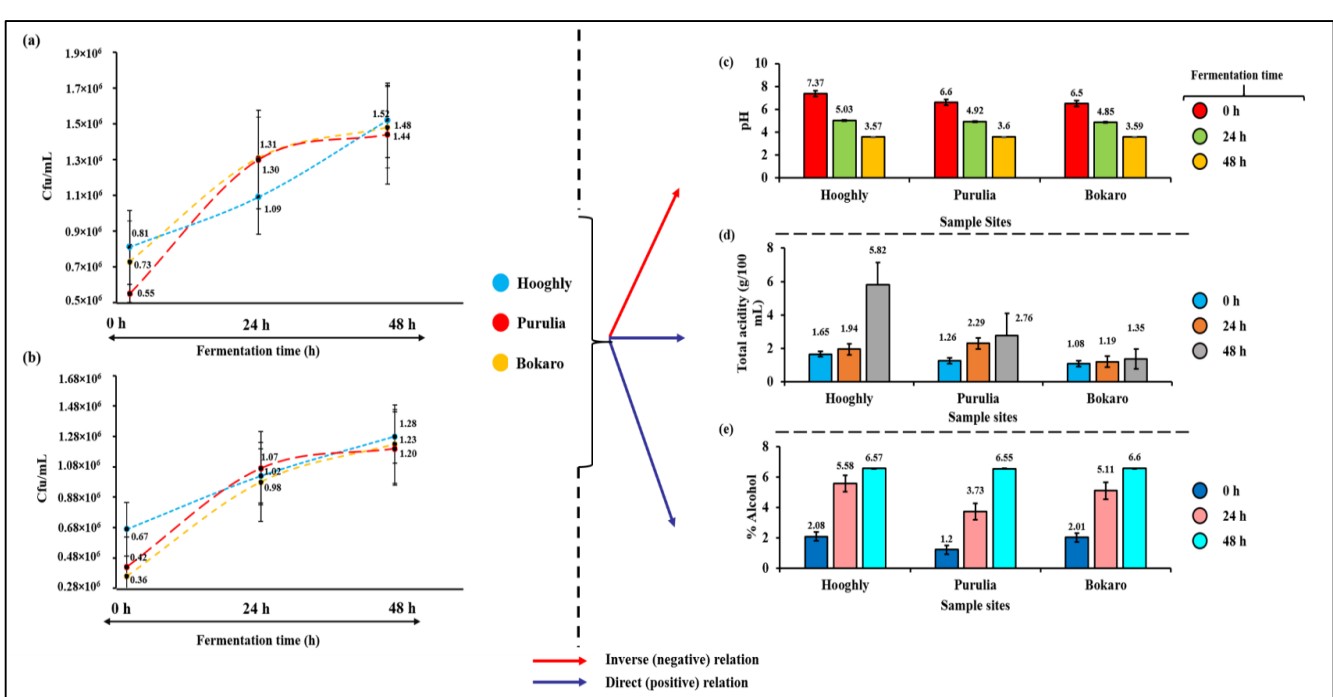

**Figure 2.** Successional changes in total microbial load and different physico-chemical parameters during the fermentation of *toddy*: (**a**) total bacterial load, (**b**) total load of yeasts, (**c**) pH, (**d**) total acidity content and (**e**) total alcohol content (error bar is representing SD).

### 3.2. Microbial Identification

A total of 50 LAB isolates were isolated primarily from the different fermentation stages of the samples collected from Hooghly, Purulia and Bokaro. Based on the growth

(A600 ≥ 1.00), 24 isolates were selected finally for further analyses. Two isolates from the fresh sap (0 h) samples of Hooghly were grouped based on the rep-PCR (3 bands: 1500 bp, 2000 bp and 2500 bp); furthermore, a single isolate (*n* = 1) was chosen as representative of the group, which was identified as *Enterococcus faecalis* THL-1 (accession number: OP968047) on the basis of 16S rRNA gene sequence (Figure 3a). Three isolates from the mid-phase of fermentation (24 h) in Hooghly samples showed a similar banding pattern (2 bands: 500 bp and 1500 bp), and the representative isolate of the group was identified as *Lacticaseibacillus paracasei* TBL-2 (OP967909) (Figure 3a). However, *Lactiplantibacillus plantarum* TPL-3 (OQ193027) was identified in the Hooghly samples from 48 h of fermentation (Figure 3a), where four isolates showed the same banding pattern with a single band at 1500 bp. The successional samples collected from Purulia displayed a reverse pattern of microbial distribution from Hooghly samples. Three isolates from 0 h fermentation of Purulia samples were found to share the same banding pattern that was observed in 24 h samples from Hooghly (2 bands: 500 bp and 1500 bp), and the representative isolate was identified as *Lacticaseibacillus paracasei* TPL-2 (OP968949). Furthermore, *Lactiplantibacillus plantarum* TBB-3 (OP967920) was also identified and recovered from the 24 h and 48 h fermented Purulia samples; a total of five isolates from these two groups were found to share the same banding pattern (1 band at 1500 bp) which was also observed in the 48 h fermented Hooghly samples. The similarity was found between the 0 h samples of Purulia and Bokaro as the 0 h of fermentation in Bokaro samples was found to be dominated by *Lacticaseibacillus paracasei* TPL-2 (OP968949). Interestingly, three isolates from 24 h fermented Bokaro samples were found to be the same group of isolates from 0 h of Hooghly samples which showed three bands in rep-PCR (1500 bp, 2000 bp and 2500 bp), where the representative isolate was also identified as *Enterococcus faecalis* THL-1 (OP968047). The final stage of fermentation in Bokaro samples also revealed the presence of *Lactiplantibacillus plantarum* TBB-3 (OP967920), which was also found common in the 48 h group of Purulia sample (Table 1).

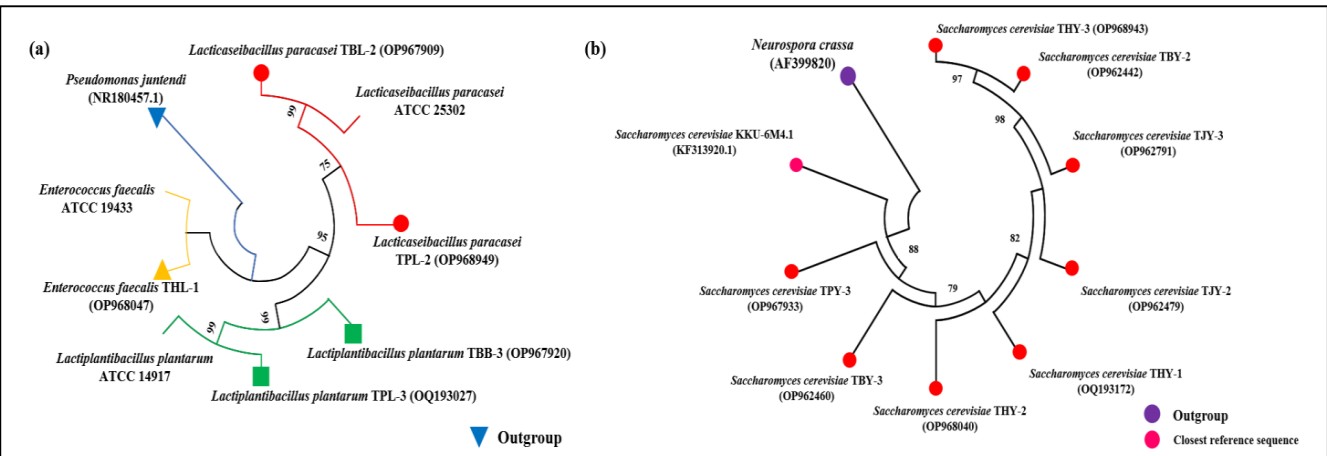

**Figure 3.** Phylogenetic tree representing the relationship between (**a**) bacteria and (**b**) yeast. The accession number is given within the parenthesis. Phylogenetic analyses of five bacterial isolates (based on 16S rRNA gene sequencing) and eight yeast isolates (based on the sequencing of the D1-D2 region of ribosomal LSU 26S rRNA gene) from *taal* and *khejur toddy* were performed using the neighbour-joining method in MEGA software v11.0.13 with *Pseudomonas juntendi* (NR180457.1) and *Neurospora crassa* (AF399820) as the outgroups for bacteria and yeasts, respectively. The number of bootstrap replications was kept default (1000). The type strains represent the closest relatives of the isolates.

**Table 1.** Identification of microorganisms in fermenting palm saps during natural fermentation of *toddy*.

| Fermentation Period (h) | *Taal Toddy* | | *Khejur Toddy* | | | |
|---|---|---|---|---|---|---|
| | Hooghly | | Purulia | | Bokaro | |
| | Bacteria | Yeast | Bacteria | Yeast | Bacteria | Yeast |
| 0 | *Enterococcus faecalis* | *Saccharomyces cerevisiae* | *Lacticaseibacillus paracasei* | *Saccharomyces cerevisiae* | *Lacticaseibacillus paracasei* | *Saccharomyces cerevisiae* |
| 24 | *Lacticaseibacillus paracasei* | *Saccharomyces cerevisiae* | *Lactiplantibacillus plantarum* | *Saccharomyces cerevisiae* | *Enterococcus faecalis* | *Saccharomyces cerevisiae* |
| 48 | *Lactiplantibacillus plantarum* | *Saccharomyces cerevisiae* | *Lactiplantibacillus plantarum* | *Saccharomyces cerevisiae* | *Lactiplantibacillus plantarum* | *Saccharomyces cerevisiae* |

Initially, 150 yeast samples were processed for flocculation test and ethanol tolerance (100 mL/L–150 mL/L). Ultimately, 26 isolates with positive flocculation and tolerance to 150 mL/L of ethanol were selected for further rep-PCR analysis. In our study, *Saccharomyces* was found to be the sole prominent yeast genus in different stages of fermentation (0–48) across all the samples (Table 1). On the basis of D1/D2 domains of large ribosomal subunit (26S rRNA gene), a single ethanol-tolerant isolate from fresh sap samples (0 h) of Hooghly was identified as *Saccharomyces cerevisiae* TPY-3 (accession number: OP967933) (Figure 3b). Three isolates from 24 h of fermentation were grouped based on a similar banding pattern (2 bands: 500 bp and 1200 bp), and the representative isolate was also identified as *Saccharomyces cerevisiae* TBY-2 (OP962442). From 48 h fermented samples, the yeast species found was *Saccharomyces cerevisiae* TBY-3 (OP962460) (Figure 3b), but its banding pattern was a little different (3 bands: 400 bp, 600 bp and 1200 bp). Two isolates from fresh sap of Purulia were found to share a similar banding pattern with fresh sap samples of Hooghly (2 bands: 500 bp and 1200 bp), and the representative isolate was identified as *Saccharomyces cerevisiae* TJY-2 (OP962479). Samples collected at 24 h and 48 h fermentation periods from Purulia showed the banding pattern: two bands at 500 bp and 1200 bp and three bands at 400 bp, 600 bp and 1200 bp, respectively, and they were also identified as *Saccharomyces cerevisiae* TJY-3 (OP962791) and *Saccharomyces cerevisiae* THY-1 (OQ193172). The isolate from the 0 h fermented Bokaro sample was found to be in the same group of 48 h Purulia samples due to the similar banding pattern. Fermentation periods of 24 h and 48 h in the Bokaro samples were also found to be enriched with *Saccharomyces cerevisiae* THY-2 (968040), where the grouped isolates showed two bands in rep-PCR (500 bp and 1200 bp).

*3.3. Physico-Chemical Changes*

A sharp decline in pH was observed in the Hooghly samples from 7.37 ± 0.12 (0 h) to 3.57 ± 0.12 (48 h). In Purulia samples, a successional decrease was observed from 6.60 ± 0.12 (0 h) to 3.63 ± 0.03 (48 h) with a pH of 4.92 ± 0.11 at mid-phase of fermentation (24 h) (Figure 2c). The pH from each succession of Bokaro samples was found to be 6.50 ± 0.24 (0 h), 4.85 ± 0.09 (24 h) and 3.59 ± 0.01 (48 h), respectively (Figure 2c). A successional change in the acidity content of fermenting samples collected from Hooghly, Purulia and Bokaro was observed. Samples collected from Hooghly showed a gradual increase in titratable acidity from 1.65 ± 0.69 g/100 mL at 0 h to 5.82 ± 0.34 g/100 L at 48 h (Figure 2d). Purulia samples also displayed a similar pattern with the varying acidity content of 1.26 ± 0.03 g/100 mL (0 h) to 2.76 ± 0.04 g/100 mL at the end of the fermentation (48 h) (Figure 2d). The total acidity content in *toddy* samples from Bokaro was found to be increased successively from 1.08 ± 0.02 g/100 mL (0 h) to 1.35 ± 0.17 g/100 mL at 48 h of fermentation (Figure 2d).

Total alcohol contents in samples of Hooghly, Purulia and Bokaro during fermentation (0 h to 48 h) increased from 2.08% ± 0.07 to 6.57% ± 0.12, 1.20% ± 0.02 to 6.55% ± 0.01

and 2.01% ± 0.02 to 6.60% ± 0.01, respectively (Figure 2e). The total alcohol content is composed of a very less proportion of methanol compared to ethanol in all the successional samples. Methanol concentration in Hooghly, Purulia and Bokaro samples was 0.38% ± 0.01, 0.5% ± 0.35 and 0.4% ± 0.01, respectively, whereas the final concentration of ethanol after the 48 h of fermentation was 6.19% ± 0.1, 6.5% ± 0.01 and 6.2% ± 0.2, respectively (Table 2). Alcohol in the fresh sap is attributed to the continuous fermentation that goes on during the daytime under sunlight, while the pots are still attached to the palm tree. A gradual decrease in °BRIX profile (Figure 4a) and total sugar content (Figure 4b) was observed throughout the natural fermentation of *toddy* from various sites. Total ester content (Figure 4c) and protein content (Figure 4d) in the fermenting samples were found to be increased during the natural fermentation. Interestingly, the sharp increase in contents of total phenolics, flavonoids and DPPH value was observed during the natural fermentation of fresh saps to *toddy* (Figure 5a–c).

**Table 2.** Successional changes in the alcohol content throughout the fermentation of *toddy* and corresponding alcoholic proportion distribution between ethanol and methanol.

| Place of Sample Collection | Successional Phase (h) | Total Alcohol % (*v/v*) | Proportion Distribution between Ethanol and Methanol | |
|---|---|---|---|---|
| | | | Ethanol (%) | Methanol (%) |
| Hooghly | 0 | 2.80 ± 0.04 | 2.65 ± 0.04 | 0.15 ± 0.20 |
| | 24 | 5.58 ± 0.07 | 5.28 ± 0.16 | 0.30 ± 0.01 |
| | 48 | 6.57 ± 0.12 | 6.19 ± 0.10 | 0.38 ± 0.00 |
| Purulia | 0 | 1.34 ± 0.02 | 1.20 ± 0.01 | 0.14 ± 0.20 |
| | 24 | 3.73 ± 0.02 | 3.36 ± 0.17 | 0.37 ± 0.02 |
| | 48 | 6.55 ± 0.01 | 6.50 ± 0.01 | 0.05 ± 0.35 |
| Bokaro | 0 | 2.01 ± 0.02 | 1.93 ± 0.02 | 0.08 ± 0.20 |
| | 24 | 5.11 ± 0.01 | 4.81 ± 0.01 | 0.30 ± 0.10 |
| | 48 | 6.60 ± 0.01 | 6.20 ± 0.20 | 0.40 ± 0.01 |

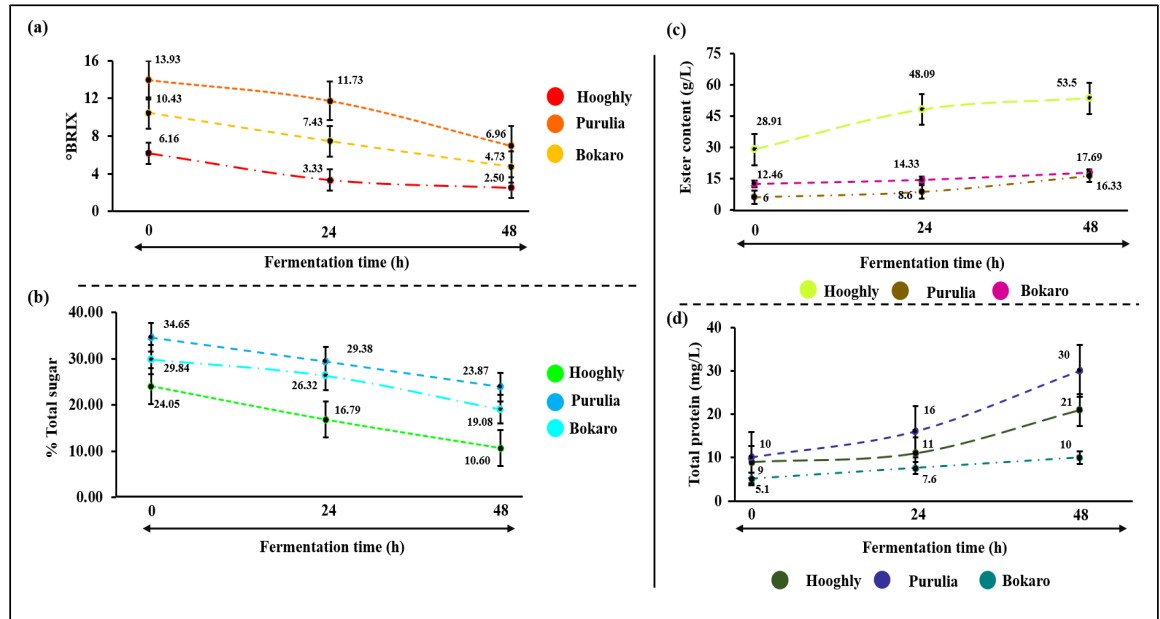

**Figure 4.** Phase-wise changes in (**a**) °BRIX content, (**b**) total sugar profile, (**c**) total ester content and (**d**) crude protein content, over the course of the natural fermentation of *toddy*.

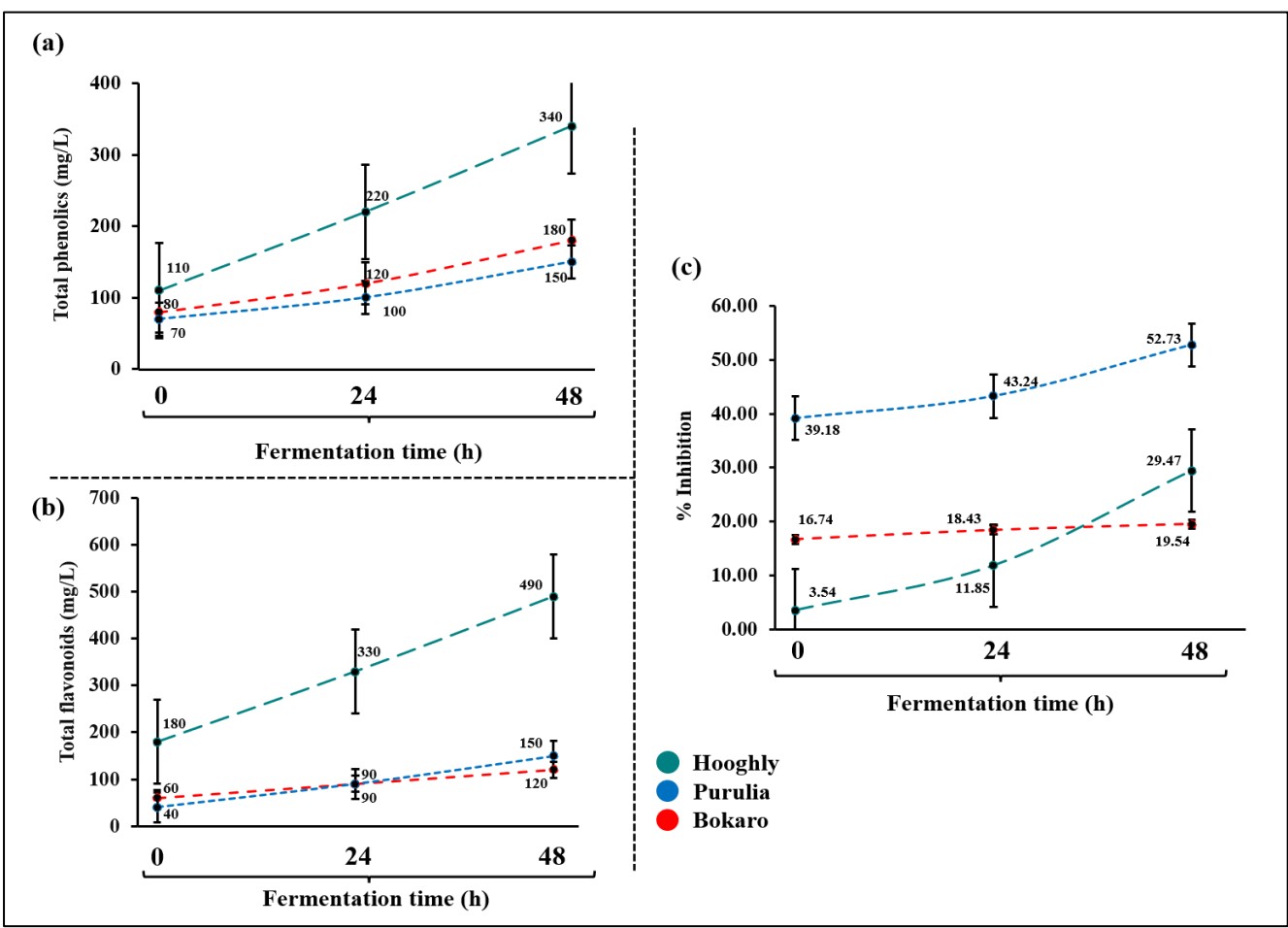

**Figure 5.** Successional increase in various bio-active and health-promoting attributes of *toddy* fermentation, viz. (**a**) content of total phenolics, (**b**) total flavonoid content and (**c**) anti-oxidant activity by radical scavenging assay.

## 4. Discussion

We hypothesise that the inter-variable dynamics play the crucial role in determining the overall fermentation and microbial kinetics in *toddy* fermentation. Hence, we studied the fermentation dynamins of two varieties of *toddy*: *taal toddy*, prepared from Palmyra palm and *khejur toddy*, prepared from date palm. Slight differences were observed between *taal toddy* and *khejur toddy* in terms of microbial compositions. Isolates from fermenting samples were grouped by the repetitive sequence-based PCR technique for discriminating the microbial strains at the species level [32], and representative isolates were identified by the 16S rRNA gene and 26S rRNA gene sequencing for bacteria and yeasts, respectively. Representative bacterial isolates from different successional rep-PCR-based groups were mostly identified as the species of LAB. *Enterococcus faecalis* was detected in fresh sap samples of Palmyra palm for *taal toddy*, whereas *Lacticaseibacillus paracasei* was detected in fresh sap samples of date palm for *khejur toddy*. The viscous liquid of fresh saps of palm tree contains sugars, soluble protein, metabolites and organic acids [33], which is a rich source of medium for supporting the growth of microorganisms, such as lactic acid bacteria, aerobic mesophilic bacteria, acetic acid bacteria and yeasts [34]. The specific role of *Enterococcus faecalis* in palm fermentation is not fully understood; however, it has been reported to have anti-microbial and other probiotic properties [35]. Similarly, species of *Lacticaseibacillus* are also known for their probiotic properties [36]. *Saccharomyces cerevisiae* was the only yeast species found from all the rep-PCR-based yeast groups throughout the fermentation. Though the presence of *Saccharomyces* during the early stages of fermentation

is quite uncommon, a similar type of observation has also been reported earlier from the successional study on oil palm drink [8]. *Lactiplantibacillus plantarum* co-existed in the final stage of fermentation in both *taal* and *khejur toddy*. *Lactiplantibacillus plantarum* improves and prolongs the shelf life [37], imparts anti-oxidant activity [38] and anti-microbial activities [39] and produces aroma, flavour [40], phenolic compounds [41] and microbial exopolysaccharides [42]. The distribution of *S. cerevisiae* across the mid-to-end phases of fermentation is correlated with the ethanol tolerance and alcohol production capability of the species [8]. Alcohol, the key metabolite in *toddy* fermentation, is mostly synthesised by *S. cerevisiae* due to its unparalleled dominance over the stages of alcoholic fermentation [8]. Besides that, *S. cerevisiae* also determines the overall quality by enhancing the bio-functional attributes along with *Lactiplantibacillus* [43]. Interestingly, we did not find any species of Acetobacteraceae from the rep-PCR-based groups; this finding can be justified by the fact that the major dominance of acetic acid bacteria has been reported mostly after 48 h of fermentation and during the storage [8,11].

Differences were also observed in the chemical profiles of the two varieties of *toddy*. Increased microbial abundance allows rapid acidification and causes the pH of the sap to decrease over the course of fermentation. Lowering pH maintains an inverse relationship with the total acidity content, due to the production of lactic acid and acetate by participating microbial communities [11]. The increase in the alcohol concentration from fresh saps to fermented end-product corroborates with the increased abundance of various species of LAB and yeasts in the final stages [44]. Interestingly, total alcohol content was composed of a maximum proportion of ethanol (6%) than methanol (<0.4%) in *toddy* samples. Alongside ethanol, it is also quite important to estimate the methanolic proportion for consumer safety purposes; though methanol is mostly toxic, its tolerance limit is up to 0.4 ($v/v$) as established by the European Union (EU) [45]. Ethanol ensures the safety of the product for human consumption if it is consumed in low quantity [46]. The ethanol content in *toddy* is <6%, which is a low-alcoholic flavoured drink, since alcoholic fermented beverage with less than 8% ethanol is classified as mild-alcoholic or low-alcoholic beverages [47]. In terms of alcohol content, *taal toddy* samples of Hooghly were found to be a little stronger than the *khejur toddy* samples of Purulia and Bokaro, which can be correlated with the comparatively strong alcoholic flavour of *taal toddy* compared with *khejur toddy*. The aroma of both types of *toddy* was sensed during the tasting and further verified by the consumer's experience. For both types of *toddy*, an astringent mouthfeel was common among the drinkers.

Total sugar content and °BRIX were found less in *taal toddy* samples compared with the *khejur toddy* samples, collected from Purulia and Bokaro, whereas, total ester content showed a reverse pattern with a higher amount in *taal toddy*. These compositional differences may be attributed to a certain number of factors, including the geographical ecology [48], palm species [49] and the difference in microbial structure [50]. A constant decrease in the °BRIX profile and content of total sugar at each successional level of fermentation may be due to microbial metabolism and growth [44,51]. Esters, the key volatile compounds behind the wine aroma, are produced through microbial action by combining acids and alcohol via the esterification reaction in different alcoholic beverages such as *baijiu* of China [52]. A successional increase in total protein content during *toddy* fermentation may be corroborated with the increased yeast dominance at the later stages of fermentation, as yeast is known as a good stabiliser and producer of protein [53].

Contents of total phenolics and total flavonoids were also found higher in *taal toddy* than in *khejur toddy*, but the differences were not statistically significant. Increased ethanol concentration aids the accumulation of flavonoids and provides stability at the later stages of fermentation, as flavonoids are more soluble in ethanol than in water [54]. Contents of phenolics and flavonoids are also attributed to the bio-nutritional enhancement of fermented beverages, since phenolic compounds are known to impart the anti-oxidant property [55]. In short, it is also noteworthy to mention that the compositional differences between the Hooghly samples and samples from Purulia and Bokaro are in line with the

fact that the variation in the species of palm tree influences the natural fermentation to a large extent [9].

## 5. Conclusions

The present study clearly explained how the spontaneous fermentation of Indian palm drink progresses and determines the product characteristics through an interrelation between the physico-chemical attributes and associated microbial populations. Accumulation of bio-active compounds (phenolics, flavonoids, etc.) in the end product supports the claim that *toddy* is a functional drink and correlates the role of microbial fermentation in synthesising some health-promoting compounds. This study also provides hints on the potential of bio-functional *Lactiplantibacillus plantarum* along with other lactic acid bacteria (*Lacticaseibacillus*, *Enterococcus*, etc.) and ethanol-producing *Saccharomyces cerevisiae* in *taal toddy* and *khejur toddy*, which may be used to develop a suitable co-culture due to their versatile bio-active and product-centric potentials. We also believe that our study provides a deep insight into a number of technical and microbiological aspects for further development of the product.

**Author Contributions:** Conceptualisation, J.P.T.; methodology, S.D.; investigation, S.D.; resources, J.P.T.; data curation, J.P.T.; writing—original draft preparation, S.D.; writing—review and editing, J.P.T.; visualisation, S.D.; supervision, J.P.T.; project administration, J.P.T.; funding acquisition, J.P.T. All authors have read and agreed to the published version of the manuscript.

**Funding:** This research received no external funding.

**Institutional Review Board Statement:** Not applicable.

**Informed Consent Statement:** Not applicable.

**Data Availability Statement:** The sequences corresponding to the isolates are available in the NCBI database with the accession numbers OP967909, OP967920, OP968047, OP968949 and OQ193027 (for bacterial isolates) and OP962442, OP962460, OP962479, OP962791, OP967933, OP968040, OP968943 and OQ193172 (for yeast isolates).

**Acknowledgments:** Jyoti P. Tamang is grateful to International Centre for Integrated Mountain Development (ICIMOD)—Mountain Chair for financial support.

**Conflicts of Interest:** The authors declare no conflict of interest.

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
