# Peer review of "Fermentation Dynamics of Naturally Fermented Palm Beverages of West Bengal and Jharkhand in India"

_fermentation, doi:10.3390/fermentation9030301_

Round 1

Reviewer 1 Report

The authors studied the fermentation profiles of naturally fermented palm beverages. Their results explained the interrelation between the physico-chemical attributes and associated microbial populations maintain the dynamics during the natural fermentation of palm drink in India. The experimental design is straightforward. The results are well-presented and the discussion part is also comprehensive. But the abstract, introduction and method sections need improvement.

Please see my detailed comments below:

Line8: suggest adding an overall conclusion of your findings.

Line 30: the intro part can be improved. the first two paragraphs described the dietary culture in India and speceis of palm trees. I think these two paragraphs can be more concise. suggest adding another paragraph to describe the importance of your study. what are the  benefits of taking fermented palm saps. sinstead of just 1-2 sentences in the last paragraph showing that this hasn't been studied before.

Line 74: more information need to show when was the two types of samples were prepared, the same day, month, or season, etc.

also, please add more details about how you prepare the containers, how to sterile the containers, etc.

line 106: what's the symbol at the begining?

Line 252: more details needed in this section. for example, what's the quality control criteria you used (score, length, etc.)?

Line 261: what reference sequences you used to do this analysis? and why?

Line 287: please specify what does the error bar mean here, standard deviation or standard error?

Line 288: did you describe the method you used to determine thses two parameters (a and b)?

Line 386: suggest adding your hypothesis in the introdiction part and have more details to support what you hypothesize like that.

Author Response

Answers to Reviewers’ Comments

Fermentation dynamics of  naturally fermented palm beverages of West Bengal and Jharkhand in India (Souvik Das and Jyoti Prakash Tamang)

Reviewer # 1

The authors studied the fermentation profiles of naturally fermented palm beverages. Their results explained the interrelation between the physico-chemical attributes and associated microbial populations maintain the dynamics during the natural fermentation of palm drink in India. The experimental design is straightforward. The results are well-presented and the discussion part is also comprehensive. But the abstract, introduction and method sections need improvement.

Please see my detailed comments below:

Line 8: suggest adding an overall conclusion of your findings.

Answer: As suggested by the Reviewer, we have added this point in the revised Abstract.

Line 30: the intro part can be improved. the first two paragraphs described the dietary culture in India and species of palm trees. I think these two paragraphs can be more concise. suggest adding another paragraph to describe the importance of your study. what are the  benefits of taking fermented palm saps. instead of just 1-2 sentences in the last paragraph showing that this hasn't been studied before.

Answer: As suggested by the Reviewer, we have addressed the above-mentioned concern in the introduction part of revised manuscript.

Line 74: more information need to show when was the two types of samples were prepared, the same day, month, or season, etc. also, please add more details about how you prepare the containers, how to sterile the containers, etc.

Answer: As per the suggestion of the Reviewer, we have included the above-mentioned details in the revised manuscript. Taal toddy samples were collected during the summer time (in the month of March-April), whereas khejur toddy samples were collected in the month of December-January (winter season). The samples were collected in the aseptic sample containers which were made sterile using the autoclave.

Line 106: what's the symbol at the beginning?

Answer: Here, Ëš symbolizes the degrees BRIX. BRIX value is usually expressed as ËšBRIX or degrees BRIX. 1Ëš BRIX generally represents 1 g of sucrose/soluble solid in 100 mL of solution (1%). As suggested by the Reviewer, we have also modified this in the revised manuscript.

Line 252: more details needed in this section. for example, what's the quality control criteria you used (score, length, etc.)?

Answer: As suggested by the Reviewer, we have added the point in the revised manuscript. Sequences with the Q score of ≥ 20-30 were processed for further analyses

Line 261: what reference sequences you used to do this analysis? and why?

Answer: Specifically, 16S rRNA database sequences were used as reference for bacterial isolates, and LSU reference database sequences from fungi type and reference material were used for the identification of yeasts. As suggested by the Reviewer, we have added this point in the revised manuscript. These two types database sequence were taken as reference because 16S rRNA gene and D1-D2 region of ribosomal LSU were used as markers for the identification of bacterial and yeast isolates, respectively.

Line 287: please specify what does the error bar mean here, standard deviation or standard error?

Answer: Here, error bar indicates the standard deviation (SD).

Line 288: did you describe the method you used to determine theses two parameters (a and b)?

Answer: Yes, we have described the method of total load count of bacteria and yeast in the first paragraph of microbiological analysis section within materials and methods. In the revised manuscript, we have highlighted that portion with yellow colour.

Line 386: suggest adding your hypothesis in the introduction part and have more details to support what you hypothesize like that.

Answer: As suggested by the reviewer, we have added the portion in the introduction part.

Reviewer 2 Report

This article presents interesting studies on the fermentation processes of beverages made from palm juice. The amount of research presented is considerable. The article may be of interest to the readers of the journal. However, the authors need to make some adjustments to the manuscript.

These include:

The purpose of the article is vague and not specific. What do the authors mean by fermentation dynamics? Do they want to establish relationships between microbial communities and changes in physico-chemical parameters. In its present form, the study looks somewhat chaotic.

The description of the research methods also needs to be clarified. For example, with what frequency the values of pH and titratable acidity were monitored. Why was the calculation for lactic acid.

Also.

Lines 75-76 How were aseptic conditions created

lines 97 What was the amount of methanol determined for? Why not higher alcohols?

The data in Table 2 and its analysis disagree. The authors indicate 0.4% methanol as the limiting concentration, while Table 2 contains drinks with higher amounts of methanol. The authors do not emphasize this

lines  434-435 the sensory analysis was not described

lines 443-444 indicated for wine. 

lines 448-458 analysis of flavonoid and polyphenol content data is critically low. Was it worthwhile to investigate these indicators?

The conclusion in the paper is also vague and not specific.

Author Response

Answers to Reviewers’ Comments

 Fermentation dynamics of  naturally fermented palm beverages of West Bengal and Jharkhand in India (Souvik Das and Jyoti Prakash Tamang)

Reviewer # 2

This article presents interesting studies on the fermentation processes of beverages made from palm juice. The amount of research presented is considerable. The article may be of interest to the readers of the journal. However, the authors need to make some adjustments to the manuscript.

These include:

The purpose of the article is vague and not specific. What do the authors mean by fermentation dynamics? Do they want to establish relationships between microbial communities and changes in physico-chemical parameters. In its present form, the study looks somewhat chaotic.

Answer: As this concern was also asked by Reviewer 1, we have included this in the introduction part of revised manuscript and highlighted with yellow.

Here, the fermentation dynamics signifies the dynamic/successional changes of different associated variables with the progress of fermentation. This dynamics ultimately determine the overall product quality by maintaining an interrelation with the microbial communities. Furthermore, this dynamics is regulated by a number of local factors which have already been mentioned in the manuscript. So, it is quite essential to study this dynamics for product characterization and consumer safety.

The description of the research methods also needs to be clarified. For example, with what frequency the values of pH and titratable acidity were monitored. Why was the calculation for lactic acid.

Answer: All the physico-chemical parameters including pH and acidity (in triplicates) were analysed in the samples from each stage/succession of fermentation, viz 0 h, 24 h and 48 h.

In our study, toddy fermentation was found to be controlled by lactic acid bacteria mostly. So, it is quite expected that lactic acid will be more prominent than others. For this reason, lactic was calculated

Lines 75-76 How were aseptic conditions created

Answer: As suggested by the Reviewer, we have re-framed the sentence as complete aseptic condition is not possible to maintain during the collection of natural samples. So, we have removed the word “aseptically”. But all the necessary precautions were taken to minimize the unwanted contamination. The sample bottles were pre-sterilized (using autoclave) and properly packed and only opened for a small amount of time prior to sample pouring. The surface, where the bottles were kept during collection, were also wiped with 70% alcohol.

Lines 97 What was the amount of methanol determined for? Why not higher alcohols?

Answer: Alongside ethanol (which is the key metabolite in toddy), we focused on methanolic proportion for the purpose of consumer safety, as methanol is mostly considered as toxic for human health. We also included our intension of methanol analysis in the revised manuscript.

Yes, we agree with the Reviewer. But, total profiling of higher alcohols (mainly responsible for aroma) needs metabolomic analysis and in this study, we mostly focused on spectrophotometric analysis. Further, we will also go for total profiling as suggested by the Reviewer in future.

The data in Table 2 and its analysis disagree. The authors indicate 0.4% methanol as the limiting concentration, while Table 2 contains drinks with higher amounts of methanol. The authors do not emphasize this

Answer: Yes, we agree with the Reviewer. In that table (Table 2), methanol concentration of only one sample (48 h of Purulia) was more than 0.4, but we did a mistake there. Total alcohol concentration of that sample is 6.55%, out of which ethanolic proportion is 6.50%. So, the methanolic proportion should be 0.05% (6.55% - 6.50%). By mistake it was written as 0.5%, instead of 0.05%. In the revised manuscript, we have corrected in Table 2.

Lines  434-435 the sensory analysis was not described

Answer: We did not perform any formal sensory analysis of the naturally fermented palm beverage. As we have mentioned, that the comparatively strong flavour (alcoholic) of taal toddy samples was due to their higher alcohol content. The aroma was sensed during the tasting and also verified with the consumer’s experience. We also changed the particular portion slightly in the revised manuscript.

Lines 443-444 indicated for wine. 

Answer: Yes, this type of esterification reaction has already been reported from the fermentation of ethnic alcoholic beverage, like baijiu of China. We have added this in the discussion portion of the revised manuscript.

Lines 448-458 analysis of flavonoid and polyphenol content data is critically low. Was it worthwhile to investigate these indicators?

Answer: Since, toddy fermentation is a natural fermentation, the bio-synthesis of any compound during the fermentation is uncontrolled and cannot be regulated/elevated by any external control. Though, the amount of total phenolics and flavonoids is low, but we think it is also important to analyse these compounds, as they confer a number of health benefits and bio-active potential of the drink. Moreover, the data from this analysis will also help to formulate the further starter culture for the drink.

The conclusion in the paper is also vague and not specific.

Answer: As suggested by the Reviewer, we revised the Conclusion in the revised manuscript..

Reviewer 3 Report

The authors studied fermentation dynamics of naturally fermented palm beverages (“toddy”). Number of colony formed units for bacteria and yeasts were determined. Repetitive sequence-based PCR as well as sequencing of full-length 16S rRNA gene and D1-D2 region of 25S rRNA gene were used for identification of isolated bacteria and yeasts, respectively. Numerous physico-chemical parameters (pH, acidity, concentrations of alchohols, sugars, flavonoids, phenolic compounds, proteins) were measured. This work is quite interesting but several points should be improved:

-calculating of CFU number was made after samples homogenization and storage at -20C. These steps can contribute to some biases due to cells can be disrupted (e.g. very low cell yield increasing during fermentation)

-authors proposed that strains which were isolated during the work dominated the communities but actually it is need to perform amplicon sequencing (V4/V3V4-region of 16S rRNA gene and ITS for bacteria and yeasts, respectively) or even metagenomic sequencing to suggest such thesis

-several methods of measurement gave results with high SD (e.g. total ester concentration, total flavonoids concentration, DPPH assay)

LL12-13: “…prepared from date palm (of West Bengal and 12 Jharkhand in India.” Please correct the parentheses (add the second one or delete)

L15: Are authors sure about 26S rRNA? Because in many references 25S rRNA is mentioned (doi: 10.1534/genetics.113.153197; doi: 10.1261/rna.078952.121)

L41: please replace “microflora” with microbiota

LL41-43: “diverse”, “diversifying” and “diversity” are used in three lines side by side (two of them in the one sentence). Please use some synonyms

L75: “Three sets of samples…” Did authors mean three samples from the one jar or from three different jars?

L92: Please decipher FSSAI abbreviation

L94: Why so unusual dimension was selected (g/100mL)?

L96: Please decipher “AOAC” abbreviation

LL130-131: Please delete “ where concentration and absorbance values were 130 placed on the X and Y axes of the curve, respectively”

LL164-165: Actually it is not clear why authors used homogenizer for samples in which they planned to count bacteria and yeast (many microbial cells can be disrupted during homogenization as well as after storage at -20C). Moreover, as I understood, authors used aerobic cultivation but probable some anaerobic (or facultatively anaerobic) microorganisms, which ferment palm sap, did not grow at aerobic conditions. And why direct observation using light microscope were not used?

L187: It is not clear why for DNA isolation from yeasts authors used valuable protocol (proteinase K treatment, phenol-choloroform extraction, isopropanol precipitation and etc) but for DNA isolation from bacteria – just lysozyme treatment and incubaction at 98C?

LL220-229: Why authors did not use sequencing of V4- and V3V4-regions of 16S rRNA to identify bacterial community and ITS – for identify yeasts diversity in the samples? Repetitive sequence-based PCR method seems outdated.

L235: Please specify PCR polymerase (or kit) as well as PCR amplifier which were used

L240: “D1-D2 region” > D1-D2 region of 25S rRNA gene

L263: Why did authors construct neighbor-joining but not maximum likelihood phylogenetic trees? It would be better due to ML is more accurate method. What model did authors use?

LL271-285: very low cell yield increasing (e.g. from 0.81 x 10^6 to 1.52 x 10^6 cfu/ml; it is increasing less than decade while usual for microbial cultures – at least twofold increase) and high cell number at fresh sap (almost million cells per ml). There are similar tendencies in other samples

Figure 2 : very high SD for acidity measurements or there is some problem with visualization (Fig. 2d). It would be better to delete it because authors already have pH dynamics

LL293-296: How “two isolates” can be named as single strain “Enterococcus faecalis THL-1 “. It is incorrect there are must be two different strains even if its 16S rRNA genes were indentical. Further down in the text there are a few more of these cases.

L307: “D1-D2 region of ribosomal LSU” > D1-D2 region of ribosomal LSU 25S rRNA

Figure3: What were support values for the trees? Bootstrap (if yes, how much replicates?)? Please provide support values on trees.

L315: “Further, 24 h and 48 h of 314 fermentations in Purulia were dominated by Lactiplantibacillus plantarum TBB-3…” How did the authors understand that these isolates were dominated?

L328: “150 mL/ltr” > 150 mL/L

LL328-329: “In this study, different stages 328 of fermentation (0 – 48) across all the samples were mostly dominated by genus Saccharomyces (Table 1)”. How did the authors understand that Saccharomyces was dominated? If they can isolate only Saccharomyces strains, it does not mean that Saccharomyces dominated. To identify dominant taxa it need to perform metabarcoding (with sequencing D1-D2 region or probably ITS) or shotgun metagenomic sequencing.

LL365-368: “…whereas the final concentration of methanol after the 48 h of fermentation was 6.19%±0.1, 6.5%±0.01 and 6.2%±0.2, respectively (Table 2)”. Methanol must be replaced with ethanol. Also there is high concentration of ethanol in fresh palm sap (1.2%-2.65%). How authors can explain this phenomenon? Is such property known for palm sap?

Author Response

Reviewer # 2

The authors studied fermentation dynamics of naturally fermented palm beverages (“toddy”). Number of colony formed units for bacteria and yeasts were determined. Repetitive sequence-based PCR as well as sequencing of full-length 16S rRNA gene and D1-D2 region of 25S rRNA gene were used for identification of isolated bacteria and yeasts, respectively. Numerous physico-chemical parameters (pH, acidity, concentrations of alcohols, sugars, flavonoids, phenolic compounds, proteins) were measured. This work is quite interesting but several points should be improved:

-calculating of CFU number was made after samples homogenization and storage at -20C. These steps can contribute to some biases due to cells can be disrupted (e.g. very low cell yield increasing during fermentation)

-authors proposed that strains which were isolated during the work dominated the communities but actually it is need to perform amplicon sequencing (V4/V3V4-region of 16S rRNA gene and ITS for bacteria and yeasts, respectively) or even metagenomic sequencing to suggest such thesis

-several methods of measurement gave results with high SD (e.g. total ester concentration, total flavonoids concentration, DPPH assay)

LL12-13: “…prepared from date palm (of West Bengal and 12 Jharkhand in India.” Please correct the parentheses (add the second one or delete)

Answer: As per the Reviewer’s suggestion, we have modified the section in the revised manuscript.

L15: Are authors sure about 26S rRNA? Because in many references 25S rRNA is mentioned (doi: 10.1534/genetics.113.153197; doi: 10.1261/rna.078952.121)

Answer: Yes, yeast isolates can be identified using the D1-D2 region of 26S rRNA gene; several recent and previous studies also reported the application of D1-D2 region of 26S rRNA gene as the marker to identify the yeast isolates. Here, we have attached some references: Lama and Tamang (2022), doi: https://doi.org/10.3390/fermentation8120664; Al-Dhabaan (2021), doi:  https://doi.org/10.1016/j.sjbs.2021.06.030.

L41: please replace “microflora” with microbiota

Answer: As instructed by the reviewer, microflora has been replaced with microbiota in the revised manuscript

LL41-43: “diverse”, “diversifying” and “diversity” are used in three lines side by side (two of them in the one sentence). Please use some synonyms

Answer: As suggested by the reviewer, we have used different synonyms, like variegating and multifarious, in the revised manuscript.

L75: “Three sets of samples…” Did authors mean three samples from the one jar or from three different jars?

Answer: We have collected the samples (in 3 sets) from a single earthen pot from each of the fermentation stages.

L92: Please decipher “FSSAI” abbreviation

Answer: As suggested by the Reviewer, we have added the abbreviation of FSSAI in the revised manuscript (FSSAI stands for ‘Food Safety and Standards Authority of India’).

L94: Why so unusual dimension was selected (g/100mL)?

Answer: We already mentioned that the method of FSSAI (Food Safety and Standards Authority of India) manual was used to determine the acidity content with slight modification. Generally, acidity is expressed in grams. And the rural consumers consume the drink on an average of 100-200 mL (1-2 glass) either occasionally or regularly. For that reason, we have calculated the acidity content in g/100 mL.

L96: Please decipher “AOAC” abbreviation

Answer: As suggested by the Reviewer, we have added the abbreviation of AOAC in the revised manuscript (AOAC stands for ‘Association of Official Analytical Collaboration’).

LL130-131: Please delete “where concentration and absorbance values were 130 placed on the X and Y axes of the curve, respectively”

Answer: As suggested by the Reviewer, we have deleted that portion in the revised manuscript.

LL164-165: Actually, it is not clear why authors used homogenizer for samples in which they planned to count bacteria and yeast (many microbial cells can be disrupted during homogenization as well as after storage at -20C). Moreover, as I understood, authors used aerobic cultivation but probable some anaerobic (or facultatively anaerobic) microorganisms, which ferment palm sap, did not grow at aerobic conditions. And why direct observation using light microscope were not used?

Answer: In our study, prior to total microbial count, we homogenized our samples for a better mixing as they were stored at -20Ëš C for few times. Here it is noteworthy to mention that the homogenization speed was kept quite low to avoid any microbial cell disruption. Anaerobic bacteria is not significant in palm wine fermentation and we also used the anaerobic jar during the isolation of lactic acid bacteria. Total microbial count analysis was a part of our bio-chemical and physico-chemical analyses, and we did not focus much on bacterial cell morphology analysis or the microscopic analysis of any particular bacteria.

L187: It is not clear why for DNA isolation from yeasts authors used valuable protocol (proteinase K treatment, phenol-choloroform extraction, isopropanol precipitation and etc) but for DNA isolation from bacteria – just lysozyme treatment and incubation at 98C?

Answer: For bacterial DNA extraction we have followed the protocol of Shangpliang and Tamang (2021), doi: https://doi.org/10.1016/j.idairyj.2021.105038, and for yeast isolates, protocol of Lama and Tamang (2022), doi: https://doi.org/10.3390/fermentation8120664, has been followed. Usually, the prokaryotic DNA molecules exist is coiled loop that floats in the cytoplasm, which is comparatively easy to extract but eukaryotic DNA is more complex which resides in the nucleus. For that reason, two different protocols were adopted for bacteria and yeast. Furthermore, we also tried the lysozyme and heat treatment-based method for yeast isolates, but we didn’t achieve the satisfactory outcome.

LL220-229: Why authors did not use sequencing of V4- and V3V4-regions of 16S rRNA to identify bacterial community and ITS – for identify yeasts diversity in the samples? Repetitive sequence-based PCR method seems outdated.

Answer: Here, our main intension was to determine the successional physico-chemical and bio-chemical changes during toddy fermentation and their interrelation with the microbial community dynamics. In this study, we did not focus much on microbial and mycobial community diversity in toddy fermentation. This is the reason why we didn’t go for V3-V4 and ITS-based amplicon sequencing. We used rep-PCR just to group our isolates prior to the molecular identification based on Sanger sequencing.

L235: Please specify PCR polymerase (or kit) as well as PCR amplifier which were used

Answer: As suggested by reviewer, we have added the details of polymerase/ PCR master mix kit and thermal cycler in the revised manuscript.

L240: “D1-D2 region” > D1-D2 region of 25S rRNA gene

Answer: As mentioned by the Reviewer, we have modified the line in the revised manuscript.

L263: Why did authors construct neighbor-joining but not maximum likelihood phylogenetic trees? It would be better due to ML is more accurate method. What model did authors use?

Answer: Yes, we completely agree with the reviewer. But, in this study we have used NJ approach as some recent studies also used this model while constructing the phylogenetic trees. Moreover, NJ method is frequently used for due to its precision when the data sets are comparatively smaller. Here, we have also attached some references: Lama and Tamang (2022), doi: https://doi.org/10.3390/fermentation8120664; Rai and Tamang (2022), doi:  https://doi.org/10.1007/s11274-021-03215-y; Shangpliang and Tamang (2021), doi: https://doi.org/10.1016/j.idairyj.2021.105038; Tamura, Nei and  Kumar (2004), doi: https://doi.org/10.1073/pnas.0404206101.

LL271-285: very low cell yield increasing (e.g. from 0.81 x 10^6 to 1.52 x 10^6 cfu/ml; it is increasing less than decade while usual for microbial cultures – at least twofold increase) and high cell number at fresh sap (almost million cells per ml). There are similar tendencies in other samples

Figure 2 : very high SD for acidity measurements or there is some problem with visualization (Fig. 2d). It would be better to delete it because authors already have pH dynamics

Answer: Yes, we also agree with the Reviewer in this regard. We can justify the phenomena/events by the fact that toddy fermentation is a natural and uncontrolled fermentation, that depends a number of factors including the type of palm species, local fermentation environment, geographical variation, climatic variation, and local/indigenous faunal ecology, mostly insect ecology.

LL293-296: How “two isolates” can be named as single strain “Enterococcus faecalis THL-1 “. It is incorrect there are must be two different strains even if its 16S rRNA genes were identical. Further down in the text there are a few more of these cases.

Answer: In this case, two isolates from 0 h samples of Hooghly were found to form a group based on their similar rep-PCR banding pattern. Ultimately, a single isolate (n=1) was taken from the group as the group representative and processed further for the molecular identification (we didn’t process both the isolates for the identification, rather only one representative isolate was chosen for further identification). In the revised manuscript, we re-framed the particular sentence for better clarification.

L307: “D1-D2 region of ribosomal LSU” > D1-D2 region of ribosomal LSU 25S rRNA

Figure 3: What were support values for the trees? Bootstrap (if yes, how much replicates?)? Please provide support values on trees.

Answer: As suggested by the reviewer, we have modified that portion in the revised manuscript.

Yes, we completely agree with the reviewer. In eukaryotes, ribosomal LSU ranges between 25S to 28S and it varies from domain to domain. D1-D2 region lies in the 5’ end of ribosomal LSU; in fungal/yeast domain, D1-D2 domain is a part of LSU 26S rRNA. We have also attached some references: Zhao et al., (2021), doi: https://doi.org/10.1038/s41598-021-83216-x; Aydin, Kustimur, Kalkanci and Duran (2019), doi: https://doi.org/10.1016/j.riam.2019.05.002; Liu, Yao, Deng, Ming and Zeng (2018), doi: https://doi.org/10.1016/j.biocontrol.2018.05.018; Navarro Rodenas, Carra and Morte (2018), doi: https://doi.org/10.3389/fmicb.2018.00994.

In Figure 3: Support values for both the trees were mentioned in the revised manuscript (average 0.5 to 0.98). The default bootstrap replicate value was kept during the construction of phylogenetic tree (which is 1000 in MEGA pipeline version 11).

L315: “Further, 24 h and 48 h of 314 fermentations in Purulia were dominated by Lactiplantibacillus plantarum TBB-3…” How did the authors understand that these isolates were dominated?

Answer: We agree with the Reviewer’s concern in this regard, and we have omitted the word dominant and re-phrased the sentence in the revised manuscript.

L328: “150 mL/ltr” > 150 mL/L

Answer: As suggested by the Reviewer, we have changed that portion in the revised manuscript.

LL328-329: “In this study, different stages 328 of fermentation (0 – 48) across all the samples were mostly dominated by genus Saccharomyces (Table 1)”. How did the authors understand that Saccharomyces was dominated? If they can isolate only Saccharomyces strains, it does not mean that Saccharomyces dominated. To identify dominant taxa, it need to perform metabarcoding (with sequencing D1-D2 region or probably ITS) or shotgun metagenomic sequencing.

Answer: We agree with the Reviewer’s comment on this and we have re-framed the portion in the revised manuscript. We also removed the word ‘dominant’.

As the yeasts isolates were screened primarily based on ethanol tolerance and flocculation prior to rep-PCR, it is quite expected that Saccharomyces will be more prominent than others, due to their good flocculating ability and ethanol tolerance. Moreover, Saccharomyces has also been reported as the key contributor in natural fermentation of palm wine, from several studies based on culture-dependent and culture-independent approaches.

LL365-368: “…whereas the final concentration of methanol after the 48 h of fermentation was 6.19%±0.1, 6.5%±0.01 and 6.2%±0.2, respectively (Table 2)”. Methanol must be replaced with ethanol. Also, there is high concentration of ethanol in fresh palm sap (1.2%-2.65%). How authors can explain this phenomenon? Is such property known for palm sap?

Answer: Yes, we agree with the points raised by the reviewer. We have replaced methanol with ethanol in the revised manuscript. Presence of some proportion of alcohol in the fresh sap is common; usually, the earthen pots are attached in the early morning for the accumulation of palm sap and removed in the evening or in next day morning. Alcohol in the fresh sap can be attributed to the continuous fermentation that goes on the day time under sunlight, while the pots are still attached with the palm tree. Native microbiota within the pots also contribute in the continuous fermentation to some extent.

Round 2

Reviewer 1 Report

No further comments.

Author Response

Thanks

Reviewer 2 Report

The authors corrected the manuscript and took into account all the comments. The article can be recommended for publication

Author Response

Thanks

Reviewer 3 Report

LL18-22: I did not understand what concentrations values authors give in Abstract? Are there mean values for all three toddy beverages? If yes, it is not unacceptable, especially for beverages from different palm trees. Please rewrite Abstract section.

Authors ignored my previous comment to initial version of MS “several methods of measurement gave results with high SD (e.g. total ester concentration, total flavonoids concentration, DPPH assay)”. So I decided provide several examples:

-L22: 15.79±11.81 g/L (total ester content)

-L25: 93.33±75.7 mg/L (total flavonoids concentration)

-L26: 19.82%±18.01 (antioxidant activity)

-L389: in the text authors gave following value for total acidity of toddy sample from Bokaro at 48h “1.35±0.17 g/100 mL” but in Figure2d (visualization of total acidity) SD whiskers looks much higher. Probably there is mistake in the text or in the Figure2.

It is quite high SD taking into account that all three replicates of same beverage were sampled from single jar. Probably it is caused by averaging technique (see comment LL18-22)?

L296: “The populations of bacteria and yeasts increased exponentially….”. Increasing from 0.81×106 cfu/ml to 1.52×106 cfu/ml during 48h is not like exponential growth.

Figure 3: what did authors mean “average support values for both the trees were found between 0.50 to 0.98”? Please indicate these values on the trees. And it will be useful to add closest relatives of isolated strains (e.g. species of same genus) on the trees because they are not very informative in current form.

L495: “bio-active compounds”. Authors could give examples (esters, flavonoids and etc)

L498: why did authors note only Lactiplantibacillus plantarum but not other isolated lactic acid bacteria in the Conclusions section?

Author Response

Answers to Second Round of Comments of Reviewers # 3

 Fermentation dynamics of  naturally fermented palm beverages of West Bengal and Jharkhand in India (Souvik Das and Jyoti Prakash Tamang)

Reviewer # 3

LL18-22: I did not understand what concentrations values authors give in Abstract? Are there mean values for all three toddy beverages? If yes, it is not unacceptable, especially for beverages from different palm trees. Please rewrite Abstract section.

Answer: Yes, in the Abstract portion of the previous manuscript we put the mean values, but we agree with the Reviewer and we have revised the abstract as suggested by the Reviewer.

Authors ignored my previous comment to initial version of MS “several methods of measurement gave results with high SD (e.g. total ester concentration, total flavonoids concentration, DPPH assay)”. So I decided provide several examples:

-L22: 15.79±11.81 g/L (total ester content)

Answer: Yes, we agree with the Reviewer; in the Abstract of the previous manuscript we provided the mean value of different parameters including all three toddy samples. That was the reason we got quite high range of SD values due to the averaging technique. So, as per the Reviewer suggestion we have removed that portion from the Abstract in the revised manuscript and re-phrased also.

-L25: 93.33±75.7 mg/L (total flavonoids concentration)

Answer: As mentioned in the comment of previous question, we have removed that from the Abstract portion and rephrased also.

-L26: 19.82%±18.01 (antioxidant activity)

Answer: As mentioned in the comment of previous question, we have removed that from the Abstract portion and rephrased also.

 -L389: in the text authors gave following value for total acidity of toddy sample from Bokaro at 48h “1.35±0.17 g/100 mL” but in Figure 2d (visualization of total acidity) SD whiskers looks much higher. Probably there is mistake in the text or in the Figure 2.

It is quite high SD taking into account that all three replicates of same beverage were sampled from single jar. Probably it is caused by averaging technique (see comment LL18-22)?

Answer: Yes, as suggested by the Reviewer, we have made the changes in Figure 2d in the revised manuscript.

 We agree with the Reviewer’s viewpoint; in the lines 18-22 of the Abstract portion of the previous manuscript, SD values was quite higher due to the averaging techniques, as we provided mean values including all 3 samples. In the revised manuscript, we have removed that and re-phrased also as suggested by the Rviewer.

L296: “The populations of bacteria and yeasts increased exponentially….”. Increasing from 0.81×106 cfu/ml to 1.52×106 cfu/ml during 48h is not like exponential growth.

Answer: As suggested by the Reviewer, we have omitted the word “exponentially” in the revised manuscript and replaced it with “gradually”.

Figure 3: what did authors mean “average support values for both the trees were found between 0.50 to 0.98”? Please indicate these values on the trees. And it will be useful to add closest relatives of isolated strains (e.g. species of same genus) on the trees because they are not very informative in current form.

Answer: As suggested by the Reviewer, we have revised and reformed the phylogenetic trees (Fig. 3) in the revised manuscript; we also have added the type strains (as closest relatives) of the isolates.

L495: “bio-active compounds”. Authors could give examples (esters, flavonoids and etc)

Answer: As suggested by the Reviewer, we have added that in the conclusion section of the revised manuscript.

L498: why did authors note only Lactiplantibacillus plantarum but not other isolated lactic acid bacteria in the Conclusions section?

Answer: As suggested by the Reviewer, we revised the Conclusion in the revised manuscript.